# Clarifying How Degree Entropies and Degree-Degree Correlations Relate to Network Robustness

**DOI:** 10.3390/e24091182

**Published:** 2022-08-24

**Authors:** Chris Jones, Karoline Wiesner

**Affiliations:** 1School of Mathematics, University of Bristol, Fry Building, Woodland Road, Bristol BS8 1UG, UK; 2Institut für Physik und Astronomie, Universität Potsdam, Campus Golm, Haus 28, Karl-Liebknecht-Straße 24/25, Golm, 14476 Potsdam, Germany

**Keywords:** complex networks, network robustness, degree distribution entropy, remaining degree entropy, mutual information of networks

## Abstract

It is often claimed that the entropy of a network’s degree distribution is a proxy for its robustness. Here, we clarify the link between degree distribution entropy and giant component robustness to node removal by showing that the former merely sets a lower bound to the latter for randomly configured networks when no other network characteristics are specified. Furthermore, we show that, for networks of fixed expected degree that follow degree distributions of the same form, the degree distribution entropy is not indicative of robustness. By contrast, we show that the remaining degree entropy and robustness have a positive monotonic relationship and give an analytic expression for the remaining degree entropy of the log-normal distribution. We also show that degree-degree correlations are not by themselves indicative of a network’s robustness for real networks. We propose an adjustment to how mutual information is measured which better encapsulates structural properties related to robustness.

## 1. Introduction

Complex networks are large structures of interrelated objects often found in the real world, with examples found throughout various scientific fields such as biology [1], ecology [2], sociology [3] and economics [4]. A large body of work is dedicated to measuring and understanding the properties of complex networks [5] using mathematical and computational methods. One approach to evaluating the properties of complex networks is through information theory. Information theory was introduced by Claude Shannon in his pioneering paper from 1948 [6] and is at the heart of all digital communication today, including efficient data compression. E.T. Jaynes, in his famous paper from 1959, derived equilibrium statistical mechanics from a maximisation principle of Shannon entropy [7]. The tools of statistical mechanics have been successfully used to describe and explain topology, growth, attack tolerance and other properties of complex networks (for a review, see [8]).

Information theory has been successfully applied to complex networks in a variety of ways. For example, the dynamical evolution of networks has been studied with measures based on Shannon entropy and divergence [9,10], information-theoretic frameworks based on maximum entropy of network ensembles explain the occurrence of different network models [11,12,13], as well as heterogeneity in network configurations [14], and a widely used clustering algorithm is based on efficient information compression [15].

The robustness of complex networks is of interest in many of their application areas. Several types of robustness exist, most importantly structural tolerance against random node removal and against targeted node removal (for a review, see [16]). Percolation studies have shown, for example, that scale-free networks display exceptional robustness against random node removal [17]. The metric most frequently used to measure this robustness is the critical fraction, which measures how many nodes must be removed from a network on average before it breaks apart.

In this paper, we examine the relationship between the heterogeneity of a network’s degree distribution and its robustness against random node removal. This is motivated by the observation that scale-free networks, which are said to have heterogeneous degree distributions, are highly robust against random node removal [17,18]. Additionally, the variance is a form of heterogeneity measure [19], and since the variance is related to an analytic result for the critical fraction [20], one may conclude that heterogeneity implies robustness in networks.

Wang et al. [21] define the Shannon entropy based on a network’s degree distribution as a measure of heterogeneity. They find that, under certain conditions, this entropy positively correlates with a network’s critical fraction. As a result, they conclude that heterogeneity and robustness are connected.

However, their analysis is limited in scope, as they only consider a narrow range of networks and only measure heterogeneity in one sense. We examine the relationship between robustness and degree distribution entropy for a wide variety of degree sequences from real-world networks, finding that a network’s entropy alone gives a lower bound for its critical fraction. Additionally, we compare heterogeneity to robustness for artificial networks with restricted expected degree values, using the remaining degree entropy [22] as well as degree distribution entropy. We find that heterogeneity measured by remaining degree entropy increases monotonically with critical fraction, whereas degree distribution entropy does not.

Other entropy measures may be defined on networks, such as mutual information [22]. This measures the correlation between the degree values of neighbouring nodes in a network. Correlations are often used to predict network robustness [23,24], although we show this is of limited effectiveness on real networks. We propose an adjustment to mutual information such that it measures both correlations and clustering, where clustering is a measure of common neighbours shared by adjacent nodes. We show that this mutual information with clustering is indicative of changes to robustness in ways that mutual information is not.

## 2. Background

### 2.1. Network Robustness

This paper will examine the relationship between network robustness and entropy measures. We define network robustness generally as a network’s ability to withstand damage, and in the context of this paper, we consider it to be a network’s ability to stay connected as nodes are removed. In this sense, a network’s robustness may be measured by identifying the fraction of nodes that must be removed before its “giant component” breaks apart, where the giant component is the largest connected component (LCC) in the network that is also significantly larger than any of the other components. This fraction is known as the “critical fraction”, and the larger it is, the more robust the network is said to be.

Robustness may be measured in other ways, such as examining the average size of the LCC throughout node removal [25] or how efficiently the information may be transported throughout a network [26]. However, for consistency with prior work in this area, we only consider the critical fraction. When a network undergoes node removal, nodes may be removed in a random order which models random failures or removed according to a specific ordering, corresponding to a targeted attack. The most simplistic targeting strategy is to remove higher degree nodes first, but other strategies exist such as removing nodes based on how many second neighbours they have [27] or the proportion of shortest paths which pass through a node [28].

One significant result for predicting the critical fraction for random failures is the Molloy-Reed criterion [20], which states that a randomly configured network will have a giant component if
(1)$$\frac{\langle k^{2}\rangle }{\langle k\rangle }>2,$$
where 〈k〉 is the expected degree of the network (i.e., the average degree) and $\langle k^{2}\rangle$ is the expected degree square. Note that we take “randomly configured” to be synonymous with “configured according to the configuration model”, where the configuration model is described in the [29]. The critical fraction is then given by the formula
(2)$$f_{c}=1-\frac{1}{\frac{\langle k^{2}\rangle }{\langle k\rangle }-1}.$$

This applies to a network of any given degree sequence, but only if the network is randomly configured.

If the network is non-randomly configured, the critical fraction may be computationally simulated using a method outlined by Newman and Ziff [30]. The Newman-Ziff algorithm constructs the specified network by activating one node at a time, adding edges between active nodes if they exist in the network. As each node is added the size of the LCC is recorded. By choosing some arbitrary threshold size above which the LCC is considered to be the giant component, such as 1% of the maximum possible component size, the critical fraction may be estimated as
(3)$$f_{c}=1-\frac{N_{1\%}}{N},$$
where $N_{1\%}$ is the number of active nodes when the LCC is 1% of its maximum size, and *N* is the total number of nodes in the desired network. This measures the robustness of a non-randomly configured network.

### 2.2. Heterogeneity Measures

The heterogeneity of the link distribution of networks may be measured with the degree distribution entropy and the remaining degree entropy. The degree distribution entropy of a network is defined as [21]
(4)$$H\left(p\right)=-\sum_{k=0}^{N-1}p\left(k\right)\ln\;p\left(k\right),$$
where *N* is the number of nodes in the network and p(k) is the probability of a node having degree *k*. This entropy measures the heterogeneity of the degree distribution.

However, this entropy measures heterogeneity in only one sense. While a network with maximum degree distribution entropy would have a uniform distribution of nodes of different degree values, it would have far more edges belonging to high degree nodes than edges belonging to low degree nodes. We describe this sort of heterogeneity as “node-centric”, and ask: what if one wished to measure a network’s heterogeneity in a more “edge-centric” sense? For this, we may use the remaining degree distribution and its associated entropy.

The remaining degree distribution gives the probability of landing on a node with *k* “remaining” degree when choosing an edge at random and traversing the edge in either direction with equal probability [23]. The remaining degree of a node is its degree minus one, describing the number of edges attached to the node while omitting the edge that has been traversed across. The remaining degree distribution q(k) is related to the degree distribution p(k) by the equation
(5)$$q\left(k\right)=\frac{\left(k+1)p(k+1\right)}{\langle k\rangle }.$$

Randomly choosing an edge and then traversing along it means that one is more likely to arrive at high-degree nodes than if nodes are randomly chosen. The remaining degree distribution captures this fact by providing a higher “weighting” to the probability of landing on high degree nodes than the degree distribution does, and so it provides a different perspective on the structure of a network.

The remaining degree entropy is defined as [22]
(6)$$H\left(q\right)=-\sum_{k+1=0}^{N-1}q\left(k\right)\ln\;q\left(k\right).$$

The remaining degree entropy measures the heterogeneity with respect to the distribution of edges belonging to nodes of different degree values, and consequently, the remaining degree entropy is large for networks where there are many more low degree nodes than high degree nodes. Therefore, degree distribution entropy and remaining degree entropy describe network heterogeneity in two different senses.

### 2.3. Mutual Information

Degree distribution and remaining degree entropy do not take network configuration into account, and so can only describe the properties of randomly configured networks. One aspect of network configuration which affects robustness is degree-degree correlation [23,24], and in an information-theoretic framework, this may be measured by mutual information.

In general, mutual information is defined for two random variables, *K* and $K^{\prime }$, with joint probability distribution $q(k,k^{\prime })$ and the corresponding marginal probability distributions, $q_{K}\left(k\right)$ and $q_{K^{\prime }}\left(k^{\prime }\right)$ [31]. For the purpose of this paper, $q(k,k^{\prime })$ is the joint remaining degree distribution of neighbouring nodes in a network, and the marginals are identical by construction such that $q_{K}=q_{K^{\prime }}=q$. In this case, the mutual information, $I(q\left(k\right);q\left(k^{\prime }\right))$, for a joint remaining degree distribution of neighbouring nodes $q(k,k^{\prime })$ with respective marginals q(k) and $q(k^{\prime })$ is given by
(7)$$I\left(q\left(k\right);q\left(k^{\prime }\right)\right)=\sum_{k+1=0}^{N-1}\sum_{k^{\prime }+1=0}^{N-1}q\left(k,k^{\prime }\right)\ln\left(\frac{q(k,k^{\prime })}{q\left(k\right)q\left(k^{\prime }\right)}\right),$$
where $q(k,k^{\prime })$ is the (joint) probability of an edge connecting a node of remaining degree *k* with a node of remaining degree $k^{\prime }$. Mutual information is zero when $q\left(k,k^{\prime }\right)=q\left(k\right)q\left(k^{\prime }\right)$ for all *k* and $k^{\prime }$ values, i.e., when there is no correlation between the remaining degree values of neighbouring nodes in the network. Conversely, mutual information is maximum (and takes the value $I\left(q\left(k\right);q\left(k^{\prime }\right)\right)=H\left(q\right)$) when $q(k,k^{\prime })=0$ for all $k,k^{\prime }$ pairs except one, for which $q\left(k,k^{\prime }\right)=q\left(k^{\prime }\right)$. In this case, the remaining degree values of neighbouring nodes are maximally correlated, and so nodes of remaining degree $k^{\prime }$ are connected only to nodes of remaining degree *k*.

### 2.4. Probability Distributions

It is commonly believed that most networks have a power-law degree distribution [32,33,34], and various models of network evolution produce power-law networks [35]. However, this has recently been called into question by Brodio and Clauset [36], who find that a log-normal distribution is a better descriptor of degree distribution for many networks in the real world. This is possibly because networks tend towards power-law distributions on an infinite scale, but since real-world networks are finite, they display log-normal degree distributions on smaller scales [37].

We incorporate both perspectives here by examining both power-law and log-normal degree distributions. The discrete power-law distribution has probability values described by [38]
(8)$$p\left(k\right)=\frac{k^{-\alpha }}{\sum_{n=0}^{\infty }{\left(n+k_{min}\right)}^{-\alpha }},$$
where $k_{min}$ is the minimum degree and $\sum_{n=0}^{\infty }{\left(n+k_{min}\right)}^{-\alpha }$ is a normalisation term.

The log-normal distribution has probability values given by [39]
(9)$$p\left(k\right)=\frac{1}{k\sigma \sqrt{2\pi }}\exp\left(-\frac{{\left(\ln\left(k\right)-\mu \right)}^{2}}{2\sigma^{2}}\right),$$
where μ and σ are the mean and standard deviation parameters respectively for the normal distribution that the log-normal distribution is based on. Note that for computational simulations of networks discrete distributions are used, but for analytic results we assume continuous distributions.

## 3. Results

### 3.1. The Entropy-Robustness Plane

First, we consider how degree distribution entropy and critical fraction relate to one another. To do this, we compare the Molloy-Reed critical fraction (Equation (2)) and degree distribution entropy values (Equation (4)) calculated from the degree distributions of 89 real-world networks. The data and sources for each network are provided in the Appendix A, and the results are given in Figure 1.

Note that, since only degree distributions are used, these results treat the real-world networks as if they were randomly configured. From Figure 1, we can see that the Molloy-Reed critical fraction is not a function of degree distribution entropy, although we can see that as degree distribution entropy increases, there is an increasing lower boundary for the critical fraction. The reason for this boundary is found in the maximum entropy principle. Given a critical fraction and, thus, a ratio of first and second moments, there exists a degree distribution on $N_{0}$ which maximises the entropy. This is a truncated normal distribution for the range [0,∞) [40], with a degree distribution given by the equation
(10)$$p\left(k\right)=\frac{1}{\sigma \sqrt{2\pi }Z}\exp\left(-\frac{{\left(k-\mu \right)}^{2}}{2\sigma^{2}}\right),$$
where μ and σ are the mean and standard deviation parameters of a non-truncated normal distribution and $Z=\frac{1}{2}\left(\mathrm{erf}\left(\frac{\mu }{\sigma \sqrt{2}}\right)+1\right)$ under the constraint of k∈[0,∞). The truncated normal distribution maximises entropy for a given first and second moment, however, specifying a critical fraction value only restricts the ratio between the first and second moment, and so it is necessary to identify the exact parameters which give the maximum entropy truncated normal distribution for a given critical fraction. To do this, we look for the ratio between μ and σ which satisfies these conditions. For the following derivation, we treat the truncated normal distribution as continuous.

The ratio between the first and second moments of the truncated normal distribution is given by
(11)$$\frac{\langle k^{2}\rangle }{\langle k\rangle }=\frac{\mu^{2}+\sigma^{2}+\mu \sigma \frac{\phi }{Z}}{\mu +\sigma \frac{\phi }{Z}},$$
where $\phi =\frac{1}{\sqrt{2\pi }}\exp\left(-\frac{\mu^{2}}{2\sigma^{2}}\right)$, and the entropy of the truncated normal distribution is
(12)$$H\left(p\right)=\ln\left(\sqrt{2\pi e}\sigma Z\right)-\frac{\mu \phi }{2\sigma Z}.$$

Rearranging Equations (11) and (12) for σ and equating them to one another gives
(13)$$\frac{\exp(H\left(p\right)+\frac{\mu \phi }{2\sigma Z})}{\sqrt{2\pi e}Z}=\frac{\left(\frac{\mu }{\sigma }+\frac{\phi }{Z}\right)\frac{\langle k^{2}\rangle }{\langle k\rangle }}{1+\frac{\mu^{2}}{\sigma^{2}}+\frac{\mu \phi }{\sigma Z}},$$
which may then be rearranged to give a new expression for H(p) in the form
(14)$$H\left(p\right)=\ln[\frac{\langle k^{2}\rangle }{\langle k\rangle }\sqrt{2\pi e}\left(Z\frac{\mu }{\sigma }+\phi \right)]-\ln[1+\frac{\mu^{2}}{\sigma^{2}}+\frac{\mu \phi }{\sigma Z}]-\frac{\mu \phi }{2\sigma Z},$$
where $\frac{\mu }{\sigma }$ is the only variable term when $\frac{\langle k^{2}\rangle }{\langle k\rangle }$ is held constant. Differentiating w.r.t. $\frac{\mu }{\sigma }$ and setting $\frac{\partial H}{\partial \frac{\mu }{\sigma }}=0$ gives
(15)$$0=\frac{1}{\frac{\mu }{\sigma }+\frac{\phi }{Z}}-\frac{\frac{\mu }{\sigma }+\left(1-\frac{\phi \mu }{Z\sigma }\right)\left(\frac{\mu }{\sigma }+\frac{\phi }{Z}\right)}{\frac{\phi \mu }{Z\sigma }+\frac{\mu^{2}}{\sigma^{2}}+1}+\frac{\phi }{2Z}\left(\frac{\mu^{2}}{\sigma^{2}}+\frac{\phi \mu }{Z\sigma }-1\right).$$

Equation (15) may then be solved numerically, giving
(16)$$\frac{\mu }{\sigma }\approx 0.84,$$
which is the ratio between μ and σ that maximises entropy for some value of $\frac{\langle k^{2}\rangle }{\langle k\rangle }$.

The entropy of the truncated normal distribution where $\frac{\mu }{\sigma }\approx 0.84$ is shown in Figure 1. By identifying the boundary to the ‘forbidden’ region in the entropy robustness plane, we obtain a lower bound for the critical fraction of randomly configured networks with a given degree distribution entropy value and prove that heterogeneity, as measured by degree distribution entropy, guarantees a certain amount of robustness.

As a final remark, it is worth pointing out that there is an upper bound to the robustness of randomly configured networks given by the leading eigenvalue of its non-backtracking matrix [14]. How these two bounds relate is an open question.

### 3.2. Distribution Entropies and Robustness

Second, we consider how degree distribution entropy and remaining degree entropy relate to robustness against random failures for random networks with a fixed expected degree. While previous findings suggest that degree distribution entropy is a measure of robustness [21], we find in the following that this entropy does not increase monotonically with the critical fraction. Instead, we find that the remaining degree of entropy increases monotonically with the critical fraction.

We measure the degree distribution entropy, remaining degree entropy and Molloy-Reed critical fraction for discrete power-law and log-normal distributions with 〈k〉=10 and compare entropies with critical fraction values. These results are given in Figure 2.

In order to simulate distributions that mimic finite real-world networks, we truncate their degree values from above, setting $k_{max}=1000$, and renormalise the degree distributions accordingly. For the power law distribution, this allows us to consider distributions where α>1, which better encompasses degree distributions found in the real world [36,38] than restricting α values to α∈(2,3), as done in [21].

In Figure 2a, we can see that the relationship between degree distribution entropy and critical fraction loops back on itself, so for either distribution there can be two networks with the same expected degree and degree distribution entropy that have very different critical fraction values. By contrast, in Figure 2b we see that the remaining degree entropy increases monotonically with the critical fraction.

These relationships between entropies and critical fraction cannot be shown analytically for the non-truncated power-law distribution when 1<α≤2, since the distribution’s moments are all infinite when α≤2. However, it is possible to show the relationship between entropies and critical fraction for the non-truncated continuous log-normal distribution as follows.

For the continuous log-normal distribution, the ratio between the first and second moments is given by
(17)$$\frac{\langle k^{2}\rangle }{\langle k\rangle }=\exp(\mu +\frac{3}{2}\sigma^{2}),$$
and the degree distribution entropy is
(18)$$H\left(p\right)=\frac{1}{2}+\ln\left(\sigma \sqrt{2\pi }\right)+\mu .$$

Using the fact that $\langle k\rangle =\exp\left(\mu +\frac{1}{2}\sigma^{2}\right)$ is constant, it is possible to rearrange Equations (17) and (18) to get
(19)$$\frac{\langle k^{2}\rangle }{\langle k\rangle }=\langle k\rangle \exp\left(\sigma^{2}\right),$$
(20)$$\;\;\;\;\;\;\;\;\;\;\;\;\;\;\;\;\;\;\;\;H(p)=\frac{1}{2}\left(1-\sigma^{2}\right)+\ln\left(\langle k\rangle \sigma \sqrt{2\pi }\right).$$

Both Equations (19) and (20) are functions of σ. $\frac{\langle k^{2}\rangle }{\langle k\rangle }$ (and thereby critical fraction) always increases as σ increases, and it is trivial to differentiate H(p) by σ to show that it reaches its maximum at σ=1, substantiating the turning point in degree distribution entropy shown in Figure 2a. Equation (20) is used to calculate the log-normal (theoretical) curve in Figure 2a.

To show that log-normal remaining degree entropy always increases monotonically with critical fraction, we first need the remaining degree distribution for the log-normal, given by
(21)$$q\left(k\right)=\frac{1}{\langle k\rangle \sigma \sqrt{2\pi }}\exp\left(-\frac{{\left(\ln\left(k\right)-\mu \right)}^{2}}{2\sigma^{2}}\right).$$

For mathematical consistency, the index of *k* is not increased by 1, instead the range of q(k) shifts from k∈[−1,∞−1) to k∈[0,∞). This gives remaining degree entropy of the form
(22)$$\begin{array}{cc} H(q) & =-\int_{0}^{\infty }q\left(k\right)\ln q\left(k\right)dk,\\ H(q) & =-\int_{0}^{\infty }\frac{1}{\langle k\rangle \sigma \sqrt{2\pi }}\exp\left(-\frac{{\left(\ln\left(k\right)-\mu \right)}^{2}}{2\sigma^{2}}\right)\left[-\frac{{\left(\ln\left(k\right)-\mu \right)}^{2}}{2\sigma^{2}}-\ln\left(\langle k\rangle \sigma \sqrt{2\pi }\right)\right]dk,\\ H(q) & ={\left[\int_{0}^{\infty }q\left(k\right)dk\frac{\left(\sigma^{2}+1\right)}{2}-\frac{q(k)}{2}k^{\frac{\mu }{\sigma^{2}}+1}\left(\mu -\sigma^{2}-ln\left(k\right)\right)\right]}_{0}^{\infty }+\int_{0}^{\infty }q\left(k\right)dk\;\ln\left(\langle k\rangle \sigma \sqrt{2\pi }\right), \end{array}$$
and making use of the fact that $\int_{0}^{\infty }q\left(k\right)dk=1$ and $\frac{q(k)}{2}k^{\frac{\mu }{\sigma^{2}}+1}\left(\mu -\sigma^{2}-\ln\left(k\right)\right)\to 0$ for both k→0 and k→∞ gives
(23)$$H\left(q\right)=\frac{1}{2}\left(1+\sigma^{2}\right)+\ln\left(\langle k\rangle \sigma \sqrt{2\pi }\right),$$
which always increases with σ and therefore also always increases with critical fraction when 〈k〉 is held constant. Equation (23) is used to calculate the log-normal (theoretical) curve in Figure 2b. The discrepancy between the numerical and theoretical results for the log-normal distribution occurs because the numerical results are generated using a discrete truncated log-normal distribution, whereas the theoretical results assume a non-truncated continuous distribution.

These results indicate that remaining degree entropy is a more reliable indicator of network robustness than degree distribution entropy and better captures the underlying structural factors that determine the robustness of a network. In particular, for networks with log-normal degree distributions, we have proved that remaining degree distribution increases monotonically with Molloy-Reed critical fraction.

### 3.3. Mutual Information and Robustness

The preceding results demonstrate the conditions under which degree entropies are indicative of robustness, but these results only apply to networks which are randomly configured. One information-theoretic approach to describing non-random network configuration is mutual information, which measures degree-degree correlations.

It has been suggested that degree-degree correlations are indicative of network robustness [23,24]. However, these claims rest upon the assumption that the network in question must be locally tree-like (i.e., the local neighbourhood of any given node must have a structure that is similar to a tree), and this is often not the case for real-world networks.

Orsini et al. [41] find that for various real-world networks, keeping degree-degree correlations constant while randomising other aspects of the structure is insufficient for preserving global network properties such as average betweenness centrality [42] or the average length of the shortest paths between each possible pairing of nodes. Instead, keeping the average clustering for each node degree value constant preserves the global network properties they consider. Note that “average clustering” refers to the probability of two nodes with a common neighbour also sharing an edge with one another. In the following, we demonstrate the fact that mutual information as previously defined is insufficient for measuring network structure that is relevant to robustness, and we propose an alteration to mutual information such that it captures both correlations and clustering.

First, we consider a real-world social network based on Facebook pages that mutually “like” one another, called “fb-pages-tvshow” in the Appendix A [43]. This network has its configuration altered while keeping its degree-degree correlations constant by repeatedly applying an edge swap algorithm [41]. This algorithm chooses two edges such that one node of one edge has the same degree value as one node of the other edge. The edges are then rewired by swapping the nodes with an equal degree with one another. This algorithm is depicted in Figure 3.

This edge swap algorithm is repeatedly applied to the social network, with critical fractions for random failures and targeted attacks being measured at set intervals. For targeted attacks, nodes are targeted based on degree values, simulating the scenario in which nodes are removed in descending degree value order. Critical fraction values (not to be confused with the Molloy-Reed critical fraction) are calculated using the Newman-Ziff algorithm for percolation along with Equation (3), and the results are given in Figure 4.

After each edge swap the social network’s mutual information value of $I(q\left(k\right);q\left(k^{\prime }\right))=0.7556$ remains unchanged, but as can be seen in Figure 4, critical fraction values for both random failures and targeted attacks increase significantly. This demonstrates that the mutual information, and by extension degree-degree correlations, are unable to capture all of the information about a network’s configuration that relates to robustness.

In order to both capture aspects of network configuration relevant to robustness and determine when correlations are likely to be indicative of robustness, we propose an alteration to the mutual information. The mutual information as defined in Equation (Equation 7) is dependent upon the joint probability, $q(k,k^{\prime })$, of randomly selecting an edge which connects a node of the remaining degree *k* with a node of remaining degree $k^{\prime }$. In order to incorporate information about clustering, when recording the degree values of any two nodes which share an edge, we exclude edges which go to common neighbours from either nodes’ degree value. An explanatory diagram is shown in Figure 5.

By defining the joint distribution in this way, we can measure mutual information such that it provides more information about network structure. We would expect this “mutual information with clustering” to be more indicative of robustness against node removal since adjacent nodes sharing a common neighbour (i.e., being in a triangular cluster) does not make them any more or less likely to remain connected to one another as nodes are removed from the network.

We repeat the same edge swapping procedure as before, keeping correlations constant and measuring the mutual information with clustering at set intervals. The mutual information with clustering can change as a network undergoes correlation preserving edge swaps, and the results are shown in Figure 6.

From Figure 6, we see that changes in mutual information with clustering correlate with changes in robustness against both random failures and targeted attacks, even with degree-degree correlations remaining unchanged.

## 4. Discussion

In Section 3.1, we found that a randomly configured network’s heterogeneity, as measured by the degree distribution entropy, sets the lower bound on the network’s robustness against random failures as measured by the Molloy-Reed critical fraction. This result tells us that a randomly configured network’s heterogeneity “guarantees” a certain minimum amount of robustness. However, degree distribution entropy by itself does not act particularly well as a measurement of network robustness, as it can only be used to provide a range of possible critical fraction values for a network.

Our results in Section 3.2 show that heterogeneity, as measured by remaining degree entropy, is a better measure of network robustness than heterogeneity as measured by degree distribution entropy when comparing networks of fixed expected degree. This is because the degree distribution entropy can take on different values for the same critical fraction when the expected degree is constant. By contrast, we observed that the remaining degree entropy increases monotonically for critical fraction values when the expected degree is constant. Additionally, our findings go against those of Wang et al. [21], who state that optimising the robustness of a power-law network with a fixed expected degree is the same as maximising its degree distribution entropy. This is because our methodology allows us to consider a wider range of networks. Instead, we find that optimising the robustness of a power-law (or log-normal) network is the same as maximising its *remaining degree* entropy.

One possible reason for the remaining degree entropy being more indicative of robustness than degree distribution entropy is that the degree distribution is a distribution across a somewhat arbitrarily labelled collection of nodes. The degree values of nodes act as labels for this distribution, but the degree values themselves have no impact on the probability of selecting certain nodes. Since network robustness is determined by how nodes are connected to one another, it is perhaps unsurprising that the remaining degree distribution is more relevant, since its probabilities are “weighted” by degree values. Nodes with high degree values are weighted more heavily than low degree nodes by the remaining degree distribution; the degree distribution weighs high degree and low degree nodes equally.

Finally, in Section 3.3 we can see that degree-degree correlations can fail to be indicative of robustness against random failures and targeted attacks on real networks, since they do not take clustering into account. Instead, it is possible to measure both correlations and clustering together using altered mutual information, and this is more indicative of network robustness. One consideration for further research is a more in-depth study of how best to incorporate information about both correlations and clustering into models of network robustness, as existing models only include one or the other.

## 5. Conclusions

In conclusion, we have investigated several degree entropy and degree-degree correlation measures defined on networks, delineating how they should be interpreted and how they relate to network robustness. In particular, we have proven that degree distribution entropy sets the lower bound for the robustness against random failures on all randomly configured networks. Additionally, we have shown that, for a fixed expected degree, a network’s heterogeneity as measured by the remaining degree entropy is a better indicator of robustness against random failures than its degree distribution entropy. Finally, we demonstrated that degree-degree correlations are not necessarily indicative of robustness, finding that measuring both correlations and clustering together is a more informative and reliable estimator of robustness.

## Figures and Tables

**Figure 1 entropy-24-01182-f001:**
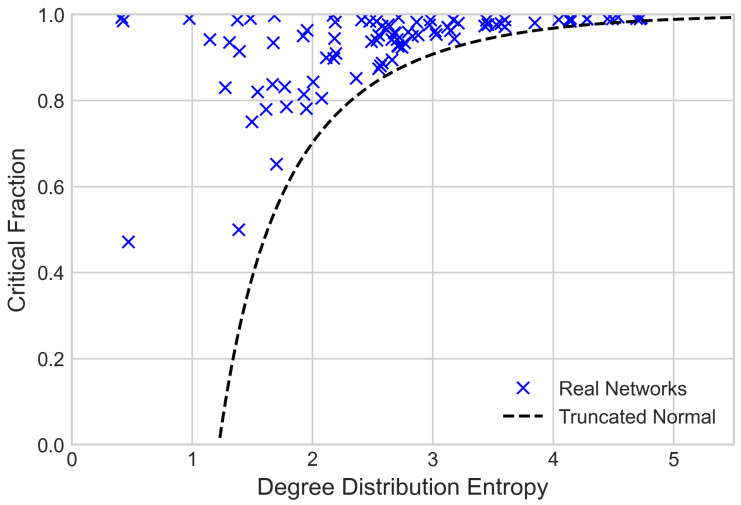
Molloy-Reed critical fraction against degree distribution entropy for real network distributions and the lower critical fraction boundary given by the truncated normal distribution.

**Figure 2 entropy-24-01182-f002:**
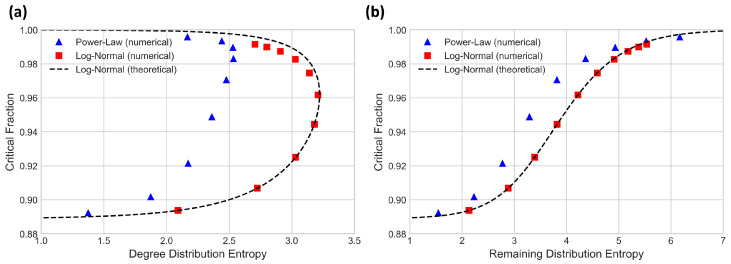
Comparison of degree entropies and critical fraction for random failures on randomly configured networks with power-law and log-normal degree distributions. In (**a**), degree distribution entropy values “loop back” on themselves for both distributions. In (**b**), remaining degree distribution increases monotonically with a critical fraction for both distributions. The theoretical curve is calculated using Equation (20) for (**a**) and Equation (23) for (**b**).

**Figure 3 entropy-24-01182-f003:**
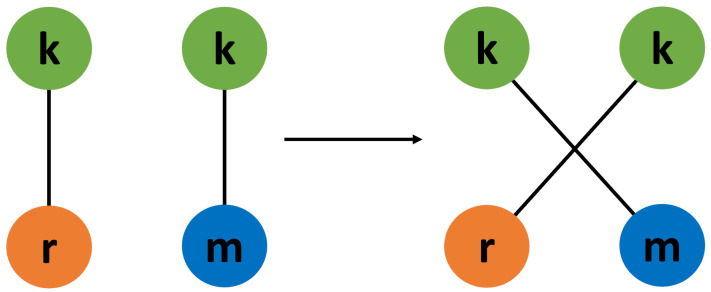
Diagram of the correlation preserving edge swap. Two edges are chosen such that they each have one node with the same degree value (in this case, the green nodes with degree k). The edge endpoints are then swapped, altering the network configuration while keeping degree-degree correlations the same.

**Figure 4 entropy-24-01182-f004:**
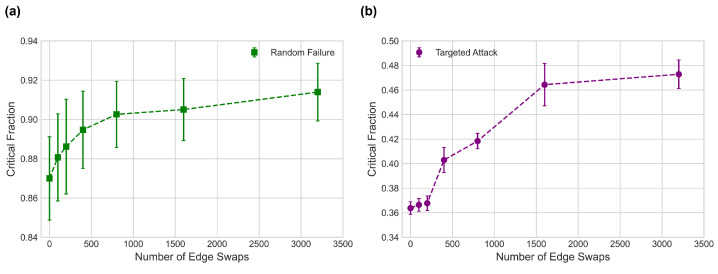
Simulated critical fraction values against number of correlation preserving edge swaps for the “fb-pages-tvshow” social network [43]. (**a**) shows data for robustness against random failures, and (**b**) is for robustness against targeted attacks. Critical fraction measurements were taken 100 times for each interval, with average values and standard deviation being recorded.

**Figure 5 entropy-24-01182-f005:**
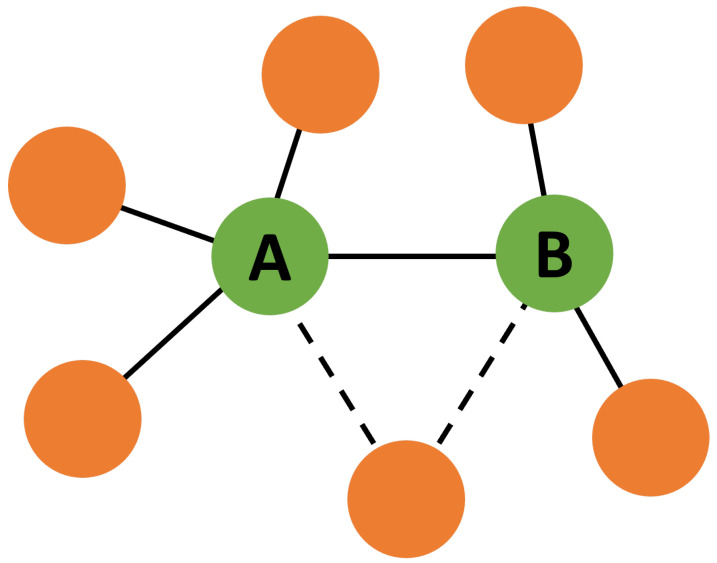
Diagram of connections for two adjacent nodes. Nodes A and B have one common neighbour with which they form a triangle, and the edges leading to that node are shown as dashed lines. For the “standard” joint distribution these edges are included when counting the degree values of A and B, but for the joint distribution “with clustering” they are not counted.

**Figure 6 entropy-24-01182-f006:**
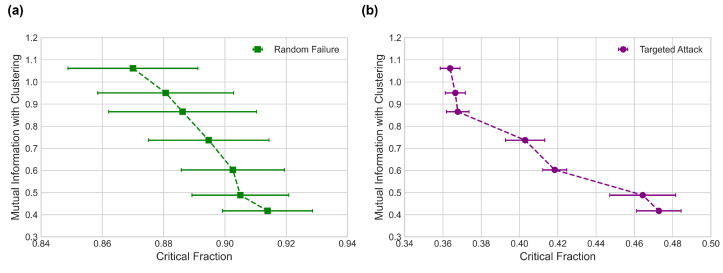
Mutual information with clustering against simulated critical fraction values the “fb-pages-tvshow” social network [43] undergoing edge swaps, for (**a**) random failures and for (**b**) targeted attacks.

## Data Availability

Not applicable.

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
