# Peer review of "Clarifying How Degree Entropies and Degree-Degree Correlations Relate to Network Robustness"

_entropy, 2022, doi:10.3390/e24091182_

Round 1

Reviewer 1 Report

The robustness of network is an important issue, which has various meanings in different scenarios for different concerns.  The paper makes two contributions about network robustness measurement. First, the remaining degree entropy is proposed to indicate the network robustness under node-removal attacks, which clearly demonstrates a better proxy than the degree distribution entropy.  Secondly, it is demonstrated that degree-degree correlations may fail to properly indicate network robustness.

The paper is overall well written. I will recommend the acceptance with two minor suggestions:

1. In the third paragraph of the ‘Discussion’ section, beside the present possible reasons, one more reason could be that, the range of entropy values depicted in Wang et al. [21] is very narrow (see Fig. 2 of [21]), and thus a “monotonical increase” is observed.  However, when a wider range is plotted (as shown in Fig. 2 of the submission), then the limitation of using entropy as the measure is clear, while the “remaining degree entropy” is a better indicator.

2. The paper focuses on the robustness measures with respect to the entropy, while other measures of network robustness may be also briefly introduced, for example, https://doi.org/10.1007/s10618-015-0447-5, and https://doi.org/10.1007/s11704-016-6108-z, as well as recent works on using machine learning for robustness estimation, e.g., https://doi.org/10.1109/TNSE.2021.3107186 .

In addition, a typo in Line 236, 'they' -> 'They'.

Author Response

We would like to thank the reviewer for their helpful comments and we have updated the manuscript based upon their recommendations.

Point 1. In the third paragraph of the ‘Discussion’ section, beside the present possible reasons, one more reason could be that, the range of entropy values depicted in Wang et al. [21] is very narrow (see Fig. 2 of [21]), and thus a “monotonical increase” is observed.  However, when a wider range is plotted (as shown in Fig. 2 of the submission), then the limitation of using entropy as the measure is clear, while the “remaining degree entropy” is a better indicator.

Response 1: We have added the sentence "This is because our methodology allows us to consider a wider range of networks." to the third paragraph of the Discussion section in order to clarify why our results differ from the findings of previous work.

Point 2. The paper focuses on the robustness measures with respect to the entropy, while other measures of network robustness may be also briefly introduced, for example, https://doi.org/10.1007/s10618-015-0447-5, and https://doi.org/10.1007/s11704-016-6108-z, as well as recent works on using machine learning for robustness estimation, e.g., https://doi.org/10.1109/TNSE.2021.3107186 .

Response 2: We have updated the second paragraph of Section 2.1 with references to other robustness measures, specifically the R measure from https://doi.org/10.1073/pnas.1009440108 and the Efficiency from https://doi.org/10.1103/physrevlett.87.198701 

Reviewer 2 Report

This work investigates several aspects relating network robustness and correlations. The paper is interesting, well written and appears to be correct. I think that it is of great interest for readers of Entropy. I have not found evident errors (only a couple of typing errors that will be corrected during editing). I think that the paper is acceptable for publication in its present form. 

P.S. I had some problems in compiling the LaTeX file for supplementary material, it should be furnished also in pdf format. 

Author Response

We would like to thank the reviewer for their feedback, and we are glad that they find the paper to be of good quality.

Reviewer 3 Report

The manuscript “Clarifying How Degree Entropies and Degree-Degree Correlations Relate to Network Robustness” investigates how distribution entropy of the node degree affects the giant component robustness to node removal in model and real complex networks.

I find the manuscript generally well written with some interesting results.

I have some revisions to address before recommending publication. On one hand, I think that the literature may be enriched (I furnish some suggestions below). On the other hand, I find some sentences not clear and need revision.

I attach the file of revisions.

Author Response

We would like to thank the reviewer for their helpful comments and we have updated the manuscript based upon their recommendations.

Point 1 : R73: There are many node attack strategies in the literature. Here I suggest mentioning other node attack strategies with references, such as betwenness, PageRank, closeness centrality, or others. I indicate some refs:
https://www.sciencedirect.com/science/article/abs/pii/S0378437114005603
https://appliednetsci.springeropen.com/articles/10.1007/s41109-021-00426-y
https://journals.plos.org/plosone/article?id=10.1371/journal.pone.0059613 

Response 1: We have updated the second paragraph of Section 2.1 to include information about other node attack strategies, adding the sentence "The most simplistic targeting strategy is to remove higher degree nodes first, but other strategies exist such as removing nodes based on how many second neighbours they have [27] or the proportion of shortest paths which pass through a node [28]." The additional references are https://doi.org/10.1016/j.physa.2014.06.079 and https://doi.org/10.1371/journal.pone.0059613 respectively.

Point 2: R88: There are other measures of network robustness bearing on the LCC notion. The widely used is the robustness R:
https://www.pnas.org/doi/full/10.1073/pnas.1009440108
https://www.nature.com/articles/s41598-019-47119-2
https://journals.plos.org/plosone/article?id=10.1371/journal.pone.0059613
https://www.nature.com/articles/s41598-018-31902-8
since this study investigates the network robustness only considering the critical fraction of nodes removed (neglecting the robustness along the removal sequence), I suggest mentioning that there are other measures of network robustness here, such as the robustness R with refs. 

Response 2: We have updated the second paragraph of Section 2.1 to mention other measures of network robustness, adding the sentences "Robustness may be measured in other ways, such as examining average size of the LCC throughout node removal [25] or how efficiently information may be transported throughout a network [26]. However, for consistency with prior work in this area we only consider the critical fraction." The new references are https://doi.org/10.1073/pnas.1009440108 and https://doi.org/10.1103/physrevlett.87.198701 respectively.

Point 3: - In Figure 2 legend I would add the network type from which the results are shown. 

Response 3: We have included information in the figure caption mentioning that these results are for randomly configured networks with the power-law and log-normal degree distributions.

Point 4: R234: The notions of ‘average betweenness centrality’ and ‘average shortest path length’ are mentioned without a definition or (at least) a reference. I suggest adding proper references here. 

Response 4: For the betweenness centrality, we have added a reference to https://doi.org/10.2307/3033543, and we have defined the average shortest path length as "the average length of shortest paths between each possible pairing of nodes"

Point 5: R236: This sentence is not clear: “they find that degree-dependent clustering is more explanatory of global network properties than e.g. correlations.”

Response 5: We have altered the third paragraph of Section 3.3 such that it now reads "Orsini et al. [41] find that for various real-world networks, keeping degree-degree correlations constant while randomising other aspects of structure is insufficient for preserving global network properties such as average betweenness centrality [42] or the average length of shortest paths between each possible pairing of nodes. Instead, keeping the average clustering for each node degree value constant preserves the global network properties they consider."

Point 6: R237: The definition is not clear to me: “Note that “clustering” refers to the probability of any three connected nodes forming a triangle.” Nodes can be connected by a path even if they are not neighbors. So this definition of clustering is not clear to me. Please explain. 

Response 6: We have updated the definition to "Note that ``average clustering'' refers to the probability of two nodes with a common neighbour also sharing an edge with one another."

Point 7: - In Figure 6 legend, I would add the name and the reference of the ‘real social network’ analyzed here

Response 7: We have now included the name and reference for the social network in paragraph 4 of Section 3.3, the caption of Figure 4 and the caption of Figure 6.